# Al^3+^ Modification of Graphene Oxide Membranes: Effect of Al Source

**DOI:** 10.3390/membranes12121237

**Published:** 2022-12-07

**Authors:** Ellen J. Robertson, Yijing Y. Stehle, Xiaoyu Hu, Luke Kilby, Katelyn Olsson, Minh Nguyen, Rebecca Cortez

**Affiliations:** 1Chemistry Department, Union College, Schenectady, NY 12308, USA; 2Department of Mechanical Engineering, Union College, Schenectady, NY 12308, USA

**Keywords:** graphene oxide, Al^3+^ modification, stability

## Abstract

Graphene oxide (GO) membranes are promising materials for water filtration applications due to abundant nanochannels in the membrane structure. Because GO membranes are unstable in water, metal cations such as Al^3+^ are often introduced to the membrane structure to promote cross-linking between individual GO sheets. Here, we describe a simple yet versatile method to incorporate Al^3+^ into GO membranes formed via a slow self-assembly process. Specifically, we directly added aluminum to acidic GO sheet solutions from a variety of sources: Al_2_O_3_, AlCl_3_ and Al foil. Each species reacts differently with water, which can affect the GO solution pH and thus the density of carboxylate groups on the sheet edges available for cross-linking to the Al^3+^ cations. We demonstrate through characterization of the GO sheet solutions as well as the as-formed membranes’ morphologies, hydrophobicities, and structures that the extent to which the Al^3+^ cross-links to the GO sheet edges vs. the GO sheet basal planes is dependent on the Al source. Our results indicate that greatest enhancements in the membrane stability occur when electrostatic and coordination interactions between Al^3+^ and the carboxylate groups on the GO sheet edges are more extensive than Al^3+^–π interactions between basal planes.

## 1. Introduction

Graphene oxide (GO) membranes are orthotropic materials that are composed of laminated GO sheets. GO membranes are promising materials for filtration and separation applications due to the abundance of nanochannels within the membrane structure, which originate from the “brick-and-mortar” lamellar stacking configuration of individual GO sheets within the membrane [1,2,3,4,5,6,7,8,9,10,11]. These individual sheets are held together through hydrogen bonding interactions between the oxygenated functional groups (OFGs) present on the edges (carboxylic acid and carboxylate groups) and basal planes (hydroxyl and epoxy groups) of the sheets, as well as van der Waals interactions between π-orbitals on the sp^2^ hybridized carbon atoms [12,13]. GO membranes are inherently unstable in water due to the net negative charge and hydrophilic nature of the GO sheets that compose the membrane [14,15]. Thus, interactions between individual sheets can be readily overcome by hydration of the OFGs present both on the membrane surface and within the membrane interior. In order to be useful materials in water filtration and separation technologies, the membranes must maintain their structural integrity in water, which is especially important if GO membrane are to be recycled and reused for multiple filtrations [16].

Previous work has shown that the GO membrane aqueous stability can be enhanced by incorporating molecular species such as ionic liquids into the membrane structure [17,18,19]. While ionic liquids can promote membrane stability through cation–π interactions, the molecular species must first be covalently attached to the graphene oxide sheet surfaces [17]. Metal ions (e.g., Na^+^, Ca^2+^, Mg^2+^, Fe^3+^, or Al^3+^) can improve membrane stability when simply incorporated into the membrane structure, with greatest stability enhancements observed with trivalent cations [6,8,11,20,21,22,23,24]. Here, electrostatic and coordination interactions between the cations and the OFGs stabilize the membrane by reducing the net negative charge of the sheets, increasing membrane hydrophobicity, and enhancing the interactions between the sheets via cross-linking. Further stabilization is imparted through cation–π interactions between the basal planes of the sheets [18]. Cation-modified GO membranes are typically formed through the fast vacuum-assisted film formation process [6,21,25]. Compared to other trivalent cations, Al^3+^ is a more versatile and low-cost cation to incorporate into GO membranes, producing Al^3+^-modified GO (AGO) membranes with exceptional stability in an aqueous solution [11,25]. Previous studies have shown that AGO membranes can persist in water for several weeks, leaching GO sheets to an extent less than 0.1% [11]. This enhanced stability indicates strong interactions between the GO sheets and Al^3+^ cations that likely minimize leaching of the GO sheets and Al^3+^ cations from the membrane. Al^3+^ can be introduced to the GO membrane from a variety of readily available Al-containing sources (alumina, aluminum salts, and aluminum foil) and through a variety of different coordination methods. In the “post-coordination” method, as-prepared GO membranes are soaked in an Al^3+^ solution. This method is simple yet has the least effect on the membrane structure because the lamellar structure of the membrane is already established [8,21]. In the “in-situ” coordination method, the GO sheet solution is passed through anodized aluminum oxide (AAO) filters which release Al^3+^ cations during the filtration process. Although these membranes show enhanced stability in water, it is difficult to precisely control the amount of Al^3+^ added to the membrane [2,14,25]. Lastly, the “pre-coordination” method introduces Al^3+^ cations to the GO sheet solution before membrane formation using either aluminum salts [6,11,24] or aluminum foil [20,26]. While this method allows for fine control over the amount of Al^3+^ within the membrane, the vacuum filtration process limits the extent to which Al^3+^ coordination orients the sheets during stacking and lamination, which can limit membrane stability [5].

A more versatile and precise method for forming AGO membranes is through the slow self-assembly process, in which membranes form during the drying of a GO sheet solution [5]. Our previous work demonstrated that, unlike membranes formed via the fast vacuum filtration method, unmodified GO-membranes formed via slow evaporation-induced self-assembly were stable in an aqueous solution under static soaking conditions due to the formation of extensive OFG–OFG interactions [22]. Using this process, Al^3+^ can be introduced to the GO membrane from all three common Al-containing species mentioned above via the pre-coordination method. These AGO membranes have the potential to be structurally different from each other as well as structurally different from corresponding AGO membranes formed via pre-coordination followed by vacuum filtration. For each Al^3+^ source, it is possible that the mechanism by which Al^3+^ is released to the GO sheet solution and cross-links the GO membrane is different and can thus produce membranes with different structures. In addition, compared to the post-coordination and in-situ coordination methods, the pre-coordination method introduces Al^3+^ to the GO sheet solution before membrane formation when the GO sheets begin to self-assemble, which can allow for maximum incorporation into the membrane structure. Moreover, pre-coordination followed by evaporation-induced self-assembly allows more time for the Al^3+^ cations to interact with and cross-link individual GO sheets in solution, and can thus enhance the GO sheet stacking, lamination structure, and cross-linking. One key question left unanswered is: Does the Al^3+^ source impact structural modifications to GO membranes, and if so, how? A thorough study on the modification of the GO membrane using the three common Al^3+^ sources via pre-coordination followed by slow self-assembly will provide a clear picture on how to tune the GO membrane structure and enhance its stability in water using a simple, versatile, and precise method.

In this work, we explore the versatility of forming AGO membranes using the pre-coordination method followed by slow self-assembly. We have already shown that we can incorporate Al^3+^ to the GO membrane structure using this method by introducing aluminum foil to the acidic GO sheet solution before the self-assembly process. These AGO membranes showed improved stability under stirring and sonication in polar solvents as well as reduced permeability to polar solvent vapors [22]. The AGO membrane properties were in part attributed to a decrease in the intra-layer spacing due to cross-linking between Al^3+^ and carboxylate groups on the GO sheet edges. Here, we extend this method to both alumina and aluminum chloride. Each Al-containing species reacts differently in acidic solution, which has the potential to affect the solution pH and thus the number of negatively charged carboxylate groups on the GO sheet edges available for Al^3+^ cross-linking. Thus, it is possible that we can control the extent to which the Al^3+^ cations form intra-layer interactions between the sheet edges or inter-layer interactions between the sheet basal planes by simply adding the appropriate Al-containing source directly to a commercially available GO sheet solution. The extent to which Al^3+^ intra-layer interactions exist compared to inter-layer interactions can greatly affect the overall membrane structure and thus its stability in water.

To explore the effect of the Al^3+^ source on the GO sheet solution properties, we measured the pH and surface tension of each solution. Subsequently, we characterized the morphology and hydrophobicity of each GO and AGO membrane using optical, scanning electron, and atomic force microscopy, as well as contact angle measurements. We also characterized each membrane structure using infrared and X-ray photoelectron spectroscopy. Finally, we assessed the stabilities of the GO and AGO membranes in water under the mechanical stresses of stirring and sonication. Our results show that the Al^3+^ source affects the GO solution properties, membrane structures, and resulting membrane stabilities in water. This research can further provide valuable guidance for developing high performance GO membranes which can be applied in aqueous environments.

## 2. Materials and Methods

### 2.1. Materials and Membrane Fabrication

Graphene oxide water dispersions (4 mg/mL) were purchased from Graphenea (Cambridge, MA, USA). Alumina powder (activated, neutral, Brockmann I, Sigma Aldrich, Burlington, MA, USA), aluminum chloride hexahydrate (ReagentPlus, 99%, Sigma Aldrich, Burlington, MA, USA), Aluminum (Al) foil (0.024 cm thick, Puratronic, 99.997% purity, Alfa Aesar, Haverhill, MA, USA) were used as Al^3+^ sources. All the reagents were used as received without further purification. Solutions of the GO sheets and 0.5 mmol of the Al-containing species were prepared by simply dissolving the Al^3+^ sources in the GO solutions followed by 4 weeks of stirring. Free-standing GO and AGO membranes were obtained by an ambient evaporation drying process (21^°^C, humidity of 50~60%) in a Teflon evaporating dish. The entire drying process took about 5 days to obtain a membrane with an average thickness of 12 ± 2 µm as measured by a thickness gauge (DeFelsko, PosiTector 6000, Ogdensburg, NY, USA).

### 2.2. Solution, Membrane Structure, and Membrane Morphology Characterization

The pH values of the GO sheet solutions were continuously measured using a pH meter (Accumet, Fisher Scientific, Nazareth, PA, USA). A Biolin Scientific Optical Tensiometer (Theta Lite, Nanoscience Instruments, Phoenix, AZ, USA) using OneAttension software was utilized to determine the surface tension of the GO and AGO solutions and to measure the contact angles of the dried membranes drop casted onto glass slides. Five to ten readings for each sample were collected to ensure the results were repeatable. The surface morphologies of the as-prepared GO membranes were characterized by an Olympus BX-51 Optical Microscope (OM) and a Zeiss EVO Scanning Electron Microscope (SEM). Surface roughness values were calculated from atomic force microscope (AFM) images obtained using a Veeco Dimension V scanning probe microscope equipped with a Bruker RTESP probe. Elemental analysis was implemented to confirm the addition of Al to the membranes and to determine the membrane oxygen/carbon ratio using Energy Dispersive Spectroscopy (EDS). Infrared (FTIR) spectra of the membranes were taken using a Nicolet iS5 FTIR Spectrometer. UV-visible measurements were carried out using a Diode Array Spectrophotometer (HP-8453). The X-ray photoelectron (XRD) spectra were recorded using a Rigaku SmartLab X-ray Diffractometer with Al Kα (1486.6 eV, λ = 0.15418 nm) as the X-ray source.

### 2.3. Stability Measurements

To investigate the stabilities of the as-prepared GO and AGO membranes in an aqueous environment, the membranes were cut into about 0.5 cm × 0.5 cm pieces and soaked in 10 mL of ultrapure water at room temperature for 20 min before stirring or sonication. The same membranes in the solutions were also gently stirred using 10 mm × 3 mm stir bars at 200 rpm in regular 20 mL sample vials to avoid touching the samples during the stirring. The vigorous sonication experiment used a 60W 40kHz sonicator (Vevor, MH009S). For each stability experiment, photographs of the GO and AGO membranes in the solutions were taken by a digital camera at different time intervals to observe the integrity of the membranes.

## 3. Results and Discussion

### 3.1. GO and AGO Sheet Solution Properties

Aluminum cations can be introduced to an acidic aqueous solution from several sources, including alumina (Al_2_O_3_), aluminum chloride (AlCl_3_), and solid aluminum foil (Al foil). Each species reacts differently in water to produce aqueous Al^3+^ and corresponding aqua ions, which can affect the solution pH and thus the extent to which Al^3+^ interacts with the OFGs on the GO sheet surfaces and edges. Below, we describe the reactivity of Al_2_O_3_, AlCl_3_, and Al foil with water and the resulting effects on the pH of the GO sheet solution (Figure 1a).

Al_2_O_3_ is insoluble in water under neutral conditions. However, Al_2_O_3_ will dissolve in acidic solutions according to the following overall reaction:
(1)
$${\mathrm{Al}}_{2}O_{3}(s)+6H^{+}(\mathrm{aq})⇌2{\mathrm{Al}}^{3+}(\mathrm{aq})+3H_{2}O(l)$$


Based on Reaction 1, it is expected that as alumina dissolves in solution, the pH will increase due to the consumption of H^+^. However, as seen in Figure 1a, the average pH of the GO-Al_2_O_3_ solution (pH = 2.40) was not significantly different than the pure GO sheet solution (pH = 2.40), which suggests that the extent of Al_2_O_3_ dissolution is relatively small. This result is consistent with previous studies showing that the rate of alumina dissolution increases with decreasing pH but that the overall dissolution can be hindered by the presence of insoluble oxygen-containing species on the alumina surface that originate from the reaction of alumina with water [27].

Aluminum chloride is soluble in aqueous solutions. Upon dissolution, Al^3+^, a Lewis acid, complexes with water to form the hexaaqua ion:
(2)
$${\mathrm{Al}}^{3+}(\mathrm{aq})+6H_{2}O(l)⇌{\left[\mathrm{Al}{\left(H_{2}O\right)}_{6}\right]}^{3+}(\mathrm{aq})$$


Because Al^3+^ is a small and highly charged cation, the hexaaqua ion is acidic (pKa = 5.0) and can further react with water to produce H_3_O^+^:
(3)
$${\left[\mathrm{Al}{\left(H_{2}O\right)}_{6}\right]}^{3+}(\mathrm{aq})+H_{2}O(l)⇌{\left[{\mathrm{Al}(H_{2}O)}_{5}(\mathrm{OH})\right]}^{2+}(\mathrm{aq})+{\left[H_{2}O\right]}^{+}(\mathrm{aq})$$


As seen in Figure 1a, and consistent with the reactivity of Al^3+^ in an aqueous solution, the average pH of the GO-AlCl_3_ solution (pH = 2.30) was slightly lower than that of the pure GO sheet solution.

Aluminum foil reacts in acidic solutions via an oxidation-reduction reaction to produce Al^3+^(aq) and hydrogen gas:
(4)
$$\mathrm{Al}(s)+H^{+}(\mathrm{aq})⇌{\mathrm{Al}}^{3+}(\mathrm{aq})+H_{2}(g)$$


The consumption of H^+^(aq) during this reaction is supported by the increase in the average pH of the GO-Al(s) sheet solution (pH = 2.60) compared to that of the pure GO-sheet solution.

Overall, the differing reactivity of each aluminum-containing species with the acidic GO sheet solution affected both the pH of the solution and the number of available aluminum cations that could interact with the OFGs on the GO sheet surfaces and edges. The extent and number of these interactions can further affect the properties of the GO sheets and membranes, including sheet and membrane hydrophobicity, membrane wrinkling, and membrane stability.

We previously reported that surface tension measurements of GO sheets at the air–water interface are indicative of the extent of sheet hydrophobicity and how the addition of Al^3+^ affects the assembly of the GO sheets as the membrane dries [22,23]. Previous studies have also shown that the interfacial activity of GO sheets is highly dependent on the solution pH, and that the surface tension of a GO sheet solution decreases with decreasing pH [28,29,30]. Specifically, the surface tension of a GO sheet solution with a pH value of 1 was reported to be ~52 mN/m, while the surface tension of a GO sheet solution with a pH of 14 was reported to be ~72 mN/m [29]. These studies showed that between pH 1 and 14, the zeta potential of the GO sheets changed from approximated −4 mV to approximately −45 mV as the carboxylic acid groups on the edges of the sheets became deprotonated [29,31]. Thus, at high pH, the high negative charges on the sheets caused them to be stable in the aqueous phase such that they did not adsorb to the air–water interface. At low pH, GO sheets were hydrophobic enough to adsorb to the air–water interface.

For this study, the surface tension of the pH 2.40 GO sheet solution in the absence of Al^3+^ had an initial value of ~60 mN/m that increased to ~70 mN/m over the course of 30 min. These results indicate that the GO sheets initially adsorbed to the air–water interface, but to a lesser extent than GO sheets in a pH 1 solution. Previous studies have reported that at a pH near 2.5, the GO sheet zeta potential is approximately −25 mV [31]. Thus, compared to GO sheets in a pH 1 solution, GO sheets in a pH 2.40 solution have more deprotonated carboxylic acid groups, are more hydrophilic and thus more stable in water, and adsorb to the air–water interface to a lesser extent. The increase in surface tension to 70 mN/m over time may be due to the desorption, reorganization, and/or increased interactions between the sheets at the surface as the GO surface layer formed.

In the presence of Al_2_O_3_, the surface tension of the GO sheet solution (~76 mN/m) was higher than that of both the neat air–water interface (72 mN/m) and the pure GO sheet solution (~70 mN/m). Moreover, the surface tension did not significantly change over the course of 30 min, which suggests that in the presence of Al_2_O_3_ the GO sheets did not assemble to the air–water interface. Previous studies have shown that the assembly of alumina nanoparticles to the air–water interface resulted in an increase in the surface tension relative to that of the neat air–water interface due to van der Waals forces between adsorbed particles [32]. Here, it is thus likely that unreacted nanometer-to-micrometer sized alumina particles existed in the GO sheet solution and assembled preferentially to the air–water interface over the GO sheets.

In the presence of AlCl_3_, the surface tension of the GO sheet solution immediately decreased to a value of ~45 mN/m and subsequently increased to a value of ~54 mN/m over the course of 30 min. This result is consistent with previous work that reported the surface tension of a 1 mg/mL GO sheet solution at a pH of 1 to be ~52 mN/m. The rapid decrease in surface tension suggests that the presence of Al^3+^ in solution acted to increase the surface activity of the GO sheets by increasing their hydrophobicity, which can be attributed to a decrease in the magnitude of the sheet zeta potential that results from protonation of the carboxylate groups with decreasing pH as well as interactions between the negatively charged carboxylate groups and the Al^3+^ ions. As was seen in the GO sheet solution, the increase in surface tension over time was likely due to the desorption, reorganization, and/or increased surface interactions as the GO sheet surface layer formed. That the final surface tension of the GO sheet solution in the presence of AlCl_3_ was similar to the surface tension of a GO sheet solution at pH 1 [29] suggests a similar extent of GO sheet assembly to the air–water interface under these different conditions.

In the presence of Al foil, the surface tension of the GO sheet solution immediately decreased to ~50 mN/m and continued to decrease over the course of 30 min to a value of ~30 mN/m. Here, the initial rapid surface tension decrease can be attributed to an increase in GO sheet hydrophobicity as Al^3+^ interacted with the negatively charged OFGs. The low surface tension value indicates that GO sheets in the pH 2.55 Al foil sample packed together more efficiently at the air–water interface than GO sheets in the pH 2.25 AlCl_3_ sample. Previous studies have shown that between a pH of 2 and 3, the GO sheet zeta potential changed from approximately −20 mV to −30 mV due to carboxylic acid deprotonation [31]. Thus, there are more available carboxylate groups to interact with Al^3+^ cations on GO sheets in the Al foil sample than on sheets in the AlCl_3_ sample. Our previous work showed that interactions between Al^3+^ and the carboxylate groups on the GO sheet edges act to decrease the intralayer spacing between sheets in the as-formed AGO membranes [22,23]. Here, we believe that a greater extent of interactions between Al^3+^ and the GO sheets in the Al foil samples allowed these sheets to pack together to a greater extent at the air–water interface than GO sheets in the AlCl_3_ sample.

### 3.2. GO and AGO Membrane Morphology and Structure

The differences in pH and surface tension behavior for each Al–GO sheet solution indicate that there may be differences in the morphologies and structures of GO membranes self-assembled from these different sheet solutions. We thus imaged the surfaces of the as-formed GO membranes using digital photography (Figure 2a), OM (Figure 2b), and SEM (Figure 2c). The membranes in Figure 2a,b were formed by cast-drying the GO sheet solutions in Teflon evaporation dishes. At the macroscopic scale, the unmodified GO membrane surface looked very similar to all three AGO membrane surfaces. However, the diameter of the AGO (Al foil) membrane was slightly smaller than the diameter of the Teflon dish, whereas the diameters of the unmodified GO, AGO (Al_2_O_3_), and AGO (AlCl_3_) membranes were roughly the same as the Teflon dishes. These images suggest that the AGO (Al foil) membrane shrank slightly during membrane drying, whereas the other membranes did not shrink. The shrinking of the AGO membrane may be due to extensive cross-linking between Al^3+^ and the carboxylate groups on the GO sheets edges, as suggested by the surface tension measurements.

To characterize the morphologies of the membranes at the micrometer scales, OM images (Figure 2b) were obtained of the membranes shown in Figure 2a. These images reveal the presence of wrinkles on all membrane surfaces. The wrinkles on the membrane surfaces have been explained as ‘peak and valley’ undulations of laminated GO sheets caused by the mixed interactions between the faces and edges of the GO sheets [33]. Stronger interaction between Al^3+^ and GO sheets will increase the ‘peak and valley’ undulations, and thus cause more wrinkles. Both coordination and electrostatic interactions between Al^3+^ and GO sheets in the solution hold the GO sheets together during the assembly process, which can cause more lateral sheet overlap and thus more wrinkles. Here, the addition of Al^3+^ through Al_2_O_3_ did not have an obvious effect on the density of wrinkles on the GO membrane surface. This insignificant effect is likely because Al_2_O_3_ did not produce Al^3+^ cations to a large extent when dissolved in the GO solution, as supported by the pH and surface tension data. For GO membranes formed with Al^3+^ added through AlCl_3_ and Al foil, the obvious increase in the number of wrinkles per area was likely due to the strong interactions between Al^3+^ and the individual GO sheets in the membranes.

The SEM images of GO membranes dried on silicon substrates (Figure 2c) further demonstrate that the addition of Al^3+^ from Al_2_O_3_ did not affect the extent of wrinkling, but that the addition of Al^3+^ from AlCl_3_ and Al foil did increase the extent of wrinkling on the GO membrane surface. These images additionally show large particles on the AGO (Al_2_O_3_) surface, which we attribute to undissolved Al_2_O_3_ particles. Moreover, both the OM and SEM images of the AGO (Al foil) membrane surface showed bubble-like features, which we attribute to the hydrogen gas evolved in Reaction 4.

To better understand how the extent of membrane wrinkling affected the hydrophobicity and thus wettability of the different GO membranes, we quantified the wrinkle density using average surface roughness values obtained from AFM analysis of membranes dried on silicon wafers (Figure 3a and Appendix A). Subsequently, we collected contact angle data for the corresponding membranes dried on glass slides (Figure 3b). Figure 3a shows both the average roughness (Ra) and root mean squared surface roughness (Rq) values for the membranes, which were collected from five 5 µm × 5 µm AFM scans from each membrane sample. Here, the AGO (Al_2_O_3_) membrane Ra (23.9 nm) and Rq (29.7 nm) values were on average lower than the unmodified GO membrane Ra (29.8 nm) and Rq (36.9 nm) values. Thus, adding Al_2_O_3_ to the GO sheet solution resulted in self-assembled GO membranes that had on average fewer wrinkles than unmodified GO membranes. We attribute this wrinkle decrease to the presence of Al_2_O_3_ particles that interrupted the interactions between GO sheets that lead to wrinkling.

However, both the AGO (AlCl_3_) membrane Ra (41.1 nm) and Rq (49.4 nm) values and AGO (Al foil) membrane Ra (37.6 nm) and Rq (46.2 nm) values were on average larger than the unmodified GO membrane Ra and Rq values, thus indicating an increase in the extent of membrane wrinkling, as seen in the corresponding OM and SEM images, due to strong interactions between Al^3+^ and the OFGs on the GO sheet edges. Overall, the AGO (AlCl_3_) membrane had on average the highest Ra and Rq values, and thus the greatest extent of wrinkling of all membranes studied.

To determine if the membrane surface roughness was correlated with the membrane wettability, the contact angles of water droplets on the membrane surfaces were measured (Figure 3b). These data show that the contact angle value of water on the unmodified GO membrane was 21.5°, while the contact angles increased to 27.3°, 39.6°, and 46.2° for the AGO (Al_2_O_3_), AGO (AlCl_3_) and AGO (Al foil) membranes, respectively. Thus, all AGO membranes studied were more hydrophobic than the unmodified GO membrane. For the AGO (Al_2_O_3_) membrane, this result is inconsistent with the surface roughness result, which suggests that the decrease in membrane roughness should lead to an increase in membrane wettability and thus decrease in contact angle value. We suspect that the increase in contact angle value and thus membrane hydrophobicity was in part due to screening of the negatively charged OFGs by positively charged Al_2_O_3_ particles on the membrane surface.

For the AGO (AlCl_3_) and AGO (Al foil) membranes, the increase in the contact angle values relative to that of the unmodified GO membrane is consistent with the increase in surface roughness values. However, the surface roughness data suggest that there was more extensive wrinkling in AGO (AlCl_3_) membranes than in AGO (Al foil) membranes. With this result, it is expected that the AGO (Al foil) membrane should be more wettable than the AGO (AlCl_3_) membrane. However, this is not the case based on the contact angle data, which indicate that the AGO (Al foil) membrane was more hydrophobic than the AGO (AlCl_3_) membrane. As with the AGO (Al_2_O_3_) membrane, the relative hydrophobicity of the membrane cannot be completely attributed to the membrane surface roughness. Rather, the chemistry of the membrane surface layer must also be considered along with the membrane roughness.

The chemistries of the AGO (Al foil) and AGO (Al_2_O_3_) membrane surfaces can in part be deduced from the surface tension data. The surface tension data suggest that there were more extensive interactions between Al^3+^ and GO sheets at the air–water interface in the GO-Al foil solution than in the GO-AlCl_3_ solution. The extent of surface interactions can affect the structure of the membrane surface. For the AGO (Al foil) membrane, the extensive Al^3+^–GO sheet interactions that neutralize the negative charges of the OFGs present at the very top layer of the membrane likely created a very hydrophobic surface that resulted in the highest contact angle value measured. While still present, surface tension results indicate that these interactions were likely less extensive on the AGO (AlCl_3_) membrane surface, which resulted in a membrane that was more hydrophobic than the unmodified GO membrane but less hydrophobic than the AGO (Al foil) membrane. The slight difference in the extent of wrinkling between the AGO (AlCl_3_) membrane and the AGO (Al foil) membrane may be due to the differences in the membrane surface structures that occurred as the membranes dried.

We also performed elemental analysis on all membranes to both determine the relative water content in each membrane and confirm the incorporation of Al into the membranes (Figure 4). The water content, which is related to membrane wettability, was approximated by measuring oxygen/carbon (O/C) ratios from EDS maps of each membrane (Figure 4a). To verify that changes in the AGO membrane O/C ratios were not due to reduction of the GO sheets, as has been noted in previous studies [26], we obtained UV-visible spectra of each membrane (Appendix A). Here, the spectra of all membranes were nearly identical. The peak near 228 nm is attributed to the π → π* transition and the shoulder near 300 nm is attributed to the n → π* transition of the C=O bonds [31]. Previous work has shown that GO sheets can be purified with alkaline washing, which acts to remove OFGs from the GO sheet surface [31]. This purification process causes the π → π* band to red shift to 250 nm due to the restoration of sp^2^ conjugation. Based on the similarity of the AGO spectra to the GO spectrum, we conclude that the addition of Al from all sources likely did not significantly affect the O/C ratio due to the removal of OFGs.

As seen in Figure 4a, the average O/C ratio for the unmodified GO membrane was 0.53, which was not significantly different than the O/C ratio for the AGO (Al_2_O_3_) membrane (O/C ratio = 0.51) or the AGO (AlCl_3_) membrane (O/C ratio = 0.57). It is expected that adding alumina to the GO membrane would increase the O/C ratio, especially since there is evidence that the alumina did not completely react in the acidic GO solution according to Reaction 1. However, the small amount of alumina added to the 4 mg/mL GO sheet solution would not significantly change the O/C ratio based on the error associated with the EDS measurements. Thus, it is likely that both the AGO (Al_2_O_3_) and AGO (AlCl_3_) membranes had a similar water content as the unmodified GO membrane. However, the O/C ratio for the AGO (Al foil) membrane (O/C ratio = 0.46) was significantly lower than the other membranes studied. While this result may suggest partial reduction of the GO sheets due to the introduction of Al foil as noted in other studies [26], it is more likely that the water content in the AGO (Al foil) membrane was less than that in the other membranes studied. The shrinking of the AGO (Al foil) membrane due to the prevalent Al^3+^–GO interactions at the membrane surface likely allowed for more water to evaporate from the membrane structure as it dried, which resulted in a decrease in the O/C ratio for the AGO (Al foil) membrane compared to all other membranes studied. EDS mapping (Appendix A) revealed that the carbon and oxygen atoms were uniformly distributed throughout all membranes studied.

EDS measurements were also performed to determine both the percent (Figure 4b) and lateral distribution (Figure 5 and Appendix A) of aluminum in each membrane. Based on a concentration of 0.5 mM for each Al^3+^-containing species, the wt% of Al was approximately 0.6% in the GO-Al_2_O_3_ solution and 0.3% in both the GO-AlCl_3_ and GO-Al foil solutions. As seen in Figure 4b, the average Al wt% was 0.42% for the AGO (Al_2_O_3_) membrane, 0.22% for the AGO (AlCl_3_) membrane, and 0.34% for the AGO (Al foil) membrane. Based on the error associated with the balance used to weigh the compounds, it is likely that the total amount of Al added to the GO solutions from each source was incorporated into the AGO membranes. To determine how the Al was laterally distributed in each AGO membrane, EDS mapping of each membrane was performed. These maps show that for the AGO (AlCl_3_) and AGO (Al foil) membranes, the lateral distribution of Al was homogeneous (Appendix A). However, for the AGO (Al_2_O_3_) membrane, higher Al densities were observed near particle-like features on the membrane surface than in the rest of the membrane (Figure 5). The EDS mapping supports the conclusion that these particles were likely unreacted alumina particles.

To better understand how the introduction of aluminum to the GO membranes from the different Al sources affected the membrane structure, FTIR (Figure 6a) and XRD (Figure 6b) spectra were obtained for all membranes studied. FTIR spectra can provide information related to the interactions between Al^3+^ and the GO sheet OFGs and π-orbitals. The spectrum of the unmodified GO membrane was similar to previously reported spectra [6,8,11,23,34,35], and displayed the characteristic peak for the aromatic C=C stretching modes from unoxidized graphite domains (1618 cm^−1^) as well as peaks attributed to the OFGs on the sheet surfaces, including carbonyl C=O (1718 cm^−1^), carboxylate C-O (1400 cm^−1^), epoxy/ether C-O (1222 cm^−1^), and hydroxyl/alkoxide/carboxyl C-O (1035 cm^−1^) functional groups. The broad peak near 3200 cm^−1^ is attributed to the O-H stretching modes of water molecules within the membrane structures.

The spectra of the AGO membranes exhibited key features related to interactions between Al^3+^ and the GO sheets within the membranes. For one, our previous work showed that an increase in the ratio of the peak intensity due to the carboxylate groups (1400 cm^−1^) to the peak intensity due to the carbonyl groups (1718 cm^−1^) can be attributed to an increase in the GO sheet solution pH and subsequent coordination interactions between Al^3+^ and membrane carboxylate groups [22]. Here, this ratio decreased from 1.19 for the unmodified GO membrane to 1.12 and 1.18 for the AGO (Al_2_O_3_) and AGO (AlCl_3_) membranes, respectively. This result is consistent with the pH decrease resulting from reacting AlCl_3_ with water (Reactions 2 and 3) as well as insignificant coordination interactions between Al^3+^ and membrane carboxylate groups. However, the corresponding ratio increased to 1.25 for the AGO (Al foil) membrane due to carboxylic acid group deprotonation upon an increase in the GO sheet solution pH (Reaction 4) as well as coordination interactions between Al^3+^ and resulting membrane carboxylate groups.

Previous studies of metal ion-functionalized GO membranes have attributed a peak near 1591 cm^−1^ to cross-linking between the GO sheet basal planes via cation–π interactions [6]. For the AGO (AlCl_3_) membrane spectrum, a sharp peak appeared near 1570 cm^−1^ that was not as pronounced in the spectra of the other membranes. Thus, it is likely that AGO (AlCl_3_) membranes experienced the most significant Al^3+^–π interactions between the GO sheet basal planes. For the AGO (Al_2_O_3_) membrane, the FTIR spectrum did not reveal significant interactions between Al^3+^ and the GO sheet OFGs and π-orbitals. This lack of interaction is likely due to the incomplete reaction between alumina and water (Reaction 1) that limited the number of free Al^3+^ ions available to interact with the GO sheets. Overall, the FTIR results suggest that the greatest extent of interactions between Al^3+^ and the basal planes of GO sheets were present in the AGO (AlCl_3_) membrane, while the greatest extent of interactions between Al^3+^ and the carboxylate groups on the GO sheet edges were present in the AGO (Al foil) membrane.

XRD is most effective in determining the interplanar distance increase with addition of metal cations to GO membranes. The diffraction angles (2θ) shown in Figure 6b were used to determine the interlayer distances (d-spacings) of laminated sheets based on Bragg’s law [25,34]. Inserting Al^3+^ into the gallery spaces between the GO sheet basal planes increased the d-spacing from 0.77 nm for the unmodified GO membrane to 0.85 nm for the AGO (Al_2_O_3_) membrane, to 0.87 nm for the AGO (AlCl_3_) membrane, and 0.83 nm for the AGO (Al foil) membrane (Appendix A). The slight differences in d-spacing between the different AGO membranes were likely due to the extent to which the Al^3+^ cations interacted with charged carboxylate groups on the edges of the sheets vs. the extent to which the Al^3+^ cations interacted with OFGs and π orbitals on the basal planes of the sheets. As seen in the FTIR results, the smallest d-spacing increase seen in the AGO (Al foil) membrane is consistent with significant coordination between Al^3+^ and the edges of the GO sheets, while the largest d-spacing increase seen in the AGO (AlCl_3_) is consistent with significant cation–π interactions within the AGO (AlCl_3_) membrane. Even though the FTIR spectra did not reveal extensive interactions between Al^3+^ and the OFGs or π-orbitals in the AGO (Al_2_O_3_) membrane, we suspect the increase in d-spacing relative to that for the unmodified GO membrane was due to the presence of alumina particles, and potentially some Al^3+^ cations, in the membrane gallery spaces. Overall, the XRD spectra are consistent with a model in which intra-layer Al^3+^-OFGs were more prevalent in AGO (Al foil) membranes, and interlayer Al^3+^–π interactions were more prevalent in AGO (AlCl_3_) membranes.

### 3.3. GO and AGO Membrane Stability in Water

The stability of GO membranes in water is critical to their use in filtration technologies. In unmodified GO membranes, individual sheets are held together through intra- and inter-layer hydrogen bonding as well as inter-layer van der Waals interactions. Because the GO sheets hold a net negative charge and are decorated with hydrophilic OFGs, unmodified membranes are unstable in water due to electrostatic repulsions between individual sheets as well as solvation of the hydrophilic OFGs located both on the membrane surface and within its interior. Our previous study suggested that unmodified GO membranes formed via a slow self-assembly process were more stable under soaking in water than GO membranes formed via vacuum filtration methods, but readily disintegrated under mechanical stresses of stirring and sonication [22]. Our work also showed that modifying the GO membrane with Al^3+^ introduced to the solution from Al foil increased their stability in water with stirring and sonication. Here, we explore the effect of the Al^3+^ source, and ultimately the effect of the extent of inter-plane vs. intra-plane interactions, on membrane stability in water under mechanical stresses.

Here, we assessed the stabilities of all membranes under stirring and sonication. The results of this study are illustrated as digital camera photos shown in Figure 7, Figure 8 and Figure 9. All membranes completely disintegrated within 1 h of gentle stirring (no collisions between the stir bar and membranes) at 200 rpm (Figure 7). Under a frequency of 60 Hz, both the unmodified GO and AGO membranes nearly disintegrated within 30 s of sonication (Figure 8). No further changes in the membrane solutions were observed when sonicated past 30 s. Even though all membranes did not persist for long under mechanical agitation, the addition of Al^3+^ from different sources affected the membrane stability differently over the course of 1 h of stirring and 30 s of sonication. As seen in Figure 7, an almost intact unmodified GO membrane persisted up to 20 min of stirring. Addition of alumina to the GO sheet solution decreased the AGO membrane stability such that an almost intact AGO (Al_2_O_3_) membrane only persisted up to 10 min of stirring. The membrane stability was improved when either AlCl_3_ or Al foil was added to the GO sheet solution. Here, an almost intact AGO (AlCl_3_) membrane persisted up to 30 min of stirring, while an almost intact AGO (Al foil) membrane persisted up to 40 min of stirring. These results suggest that the AGO (Al foil) membrane was the most stable in water under stirring.

The sonication results show a similar trend as the stirring results (Figure 8 and Figure 9). As seen in Figure 8, the unmodified GO membrane disintegrated into a homogeneous brown solution, with a small intact membrane still existing after 30 s of sonication. The AGO (Al_2_O_3_) membrane similarly disintegrated into a homogeneous brown solution, but no visible membrane was present after 20 s of sonication. Thus, the AGO (Al_2_O_3_) was less stable than the unmodified GO membrane under sonication. Both the AGO (AlCl_3_) and the AGO (Al foil) membranes disintegrated non-uniformly, which resulted in solutions that contained visible black flakes. These black flakes are attributed to AGO sheet aggregates rather than individual reduced GO sheets. However, some key differences existed between the AGO (AlCl_3_) and AGO (Al foil) membrane solutions after sonication. Specifically, the AGO (AlCl_3_) membrane solution was brown in color, suggesting that some of the membrane uniformly disintegrated into free GO sheets under sonication, while only black flakes could be observed in the AGO (Al foil) membrane solution.

The overall relative stabilities of all the membranes studied can be further seen when comparing 10-× diluted GO sheet solutions to the corresponding membrane solutions after 60 s of sonication (Figure 9). Here, the post-sonication unmodified GO membrane solution was slightly darker in color than the 10-× diluted GO sheet solution yet was still relatively homogenous. This result suggests that the unmodified GO membrane mostly disintegrated into free GO sheets with sonication, yet some small intact membrane pieces likely remained in the solution. The AGO (Al_2_O_3_) membrane solution after sonication looked nearly identical to the 10-× diluted AGO (Al_2_O_3_) sheet solution, indicating that the membrane completely disintegrated into free GO sheets during sonication. This result is consistent with a decrease in membrane stability upon Al^3+^ modification with Al_2_O_3_. For the AGO (AlCl_3_) membrane, the post-sonication solution was a darker brown color than either the unmodified GO or AGO (Al_2_O_3_) membrane solutions yet was still relatively homogeneous. Thus, it is likely that either more or larger intact membrane pieces still existed after sonication than was observed in the unmodified GO membrane solution, which indicates that Al^3+^ modification with AlCl_3_ increased the GO membrane stability. Lastly, the AGO (Al foil) membrane solution was the least homogenous after sonication, showing black flakes consisting of relatively large intact membrane pieces. After storage, these flakes persisted, which further demonstrates that Al^3+^-modification with Al foil was the most effective in increasing the stability of GO membranes.

These results are consistent with our previous study [22], which suggested that hydrogen bonding, ion–dipole, and dipole–dipole interactions between water molecules and OFGs both on the membrane surface and interior were strong enough to overcome the hydrogen bonding and van der Waals interactions between the individual sheets within the unmodified GO membrane. Thus, the unmodified GO membrane readily disintegrated in water under mechanical stresses. Here, we have attempted to improve membrane stability by adding Al^3+^ to the membrane structure from three different sources, which can improve GO membrane stability by replacing inter-sheet hydrogen bonds and relatively weak van der Waals interactions with overall stronger cation-to-OFGs coordinate covalent bonds, cation–π interactions, and electrostatic interactions. Modification of the GO membrane via addition of Al_2_O_3_ to the sheet solution decreased the as-formed membrane stability in water under mechanical stresses. Conversely, adding either AlCl_3_ or Al foil to the sheet solution increased the as-formed membrane stability in water under mechanical stresses.

Our previous work indicated that an increase in membrane stability in water was related to an increase in membrane hydrophobicity [22]. Here, as indicated by the contact angle results, all AGO membranes were more hydrophobic than the unmodified GO membrane, but only the AGO (AlCl_3_) and AGO (Al foil) membranes were more stable in water than the unmodified GO membrane. Our previous work also showed that an increase in membrane stability resulted in a decrease in membrane permeability to water vapor [22]. We measured the permeability of each membrane to water vapor (Appendix A) and, as expected, these results showed that the permeability of the AGO (Al_2_O_3_) membrane was greater than that of the unmodified GO membrane, while the permeabilities of the AGO (AlCl_3_) and AGO (Al foil) membranes were less than that of the unmodified GO membrane. Thus, adding either AlCl_3_ or Al foil to the acidic GO sheet solution produced enough Al^3+^ to promote membrane stability and slightly reduce membrane permeability, while adding Al_2_O_3_ did not produce enough Al^3+^ to promote membrane stability or reduce membrane permeability. Even though enhanced membrane permeability is important for filtration applications, the enhanced water stability seen in our AGO (AlCl_3_) and especially AGO (Al foil) membranes may allow them to serve as recyclable and reusable filters.

We attribute the decrease in membrane stability and increase in membrane permeability with Al_2_O_3_ addition to the limited reactivity of alumina in the acidic GO solution. We suspect that many alumina particles persisted in solution and were incorporated into the GO membrane interlayer gallery spaces upon drying. Even though these positively charged particles can screen the negative charges on the AGO (Al_2_O_3_) membrane, they likely interrupted van der Waals interactions that stabilize unmodified GO membranes in water without providing the stabilizing effects of free Al^3+^ cations. Based on the FTIR, XRD, and stability data, we suspect that the stabilizing effect in the AGO (AlCl_3_) membrane is different than in the AGO (Al foil) membrane. Inter-layer and intra-layer interactions between the GO sheets and Al^3+^ cations likely existed in both membranes. However, intra-layer coordination and electrostatic interactions between Al^3+^ and the negatively charged carboxylate groups on the sheet edges were more prevalent in the AGO (Al foil) membrane, while inter-layer Al^3+^–π interactions were more prevalent in the AGO (AlCl_3_) membrane. While these different interactions resulted in an increase in membrane hydrophobicity and consequently water stability, they likely affected the membrane structure and thus the mechanism by which each membrane disintegrated in water under mechanical stresses. Overall, it appears that GO membrane modification with Al^3+^ introduced via the oxidation of Al foil resulted in membranes with the greatest stability enhancement, thus it is likely that intra-layer interactions with Al^3+^ promote membrane stability more so than inter-layer interactions with Al^3+^.

## 4. Conclusions

In conclusion, we have demonstrated a simple and versatile method for incorporating Al^3+^ into the GO membrane structure by first adding an Al-containing species to the acidic GO sheet solution and subsequently forming membranes via a slow self-assembly process. This method allows for introducing Al^3+^ to the solution from sources that react differently in an acidic solution before membrane formation, which affects the extent of both free Al^3+^ cations and negatively charged carboxylate groups available for edge-to-edge cross-linking, and ultimately, membrane stability. We observed that Al_2_O_3_ did not significantly react with water to form Al^3+^. Here, remaining alumina particles disrupted the lamellar structure and decreased the GO membrane stability. Conversely, both AlCl_3_ and Al foil did significantly react with water to form Al^3+^, but the GO-AlCl_3_ sheet solution pH decreased while the GO-Al foil sheet solution pH increased. Surface tension measurements and membrane characterization revealed that GO sheet edge-to-edge cross-linking with Al^3+^ was more prevalent in the AGO (Al foil) membrane, while Al^3+^–π interactions were more prevalent in the AGO (AlCl_3_) membrane. These differences were likely due to pH values of the GO sheet solutions, which affected the number of available carboxylate groups on the sheet edges for cross-linking with Al^3+^. Both AGO (Al foil) and AGO (AlCl_3_) membranes were more stable in water under mechanical stresses, but, overall, the AGO (Al foil) membrane was more stable than the AGO (AlCl_3_) membrane. Thus, it is likely that edge-to-edge cross-linking imparts greater membrane stability than inter-layer Al^3+^–π interactions. Our method for incorporating Al^3+^ into the GO membrane structure can thus be used to modulate the specific type of Al^3+^–GO sheet interactions by simply selecting the appropriate Al^3+^ source and dissolving it in a commercially available GO sheet solution. The results presented here have implications for designing highly stable GO membranes for water filtration and separation technologies.

## Figures and Tables

**Figure 1 membranes-12-01237-f001:**
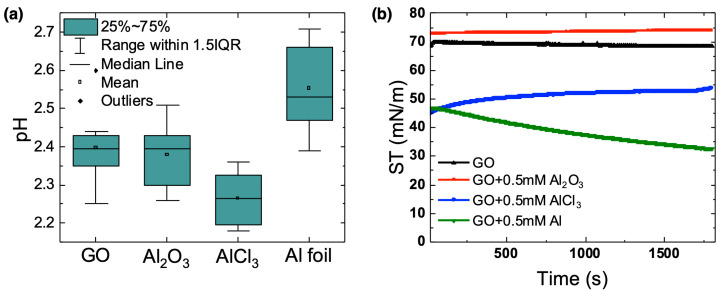
Properties of the GO and AGO sheet solutions. (**a**) pH of the GO sheet solution as a function of the Al^3+^ source added to the solution. The reported values are from at least 7 trials. (**b**) Surface tension measurements of the GO and AGO sheets solutions as a function of time. The reported values are averages of at least 5 trials.

**Figure 2 membranes-12-01237-f002:**
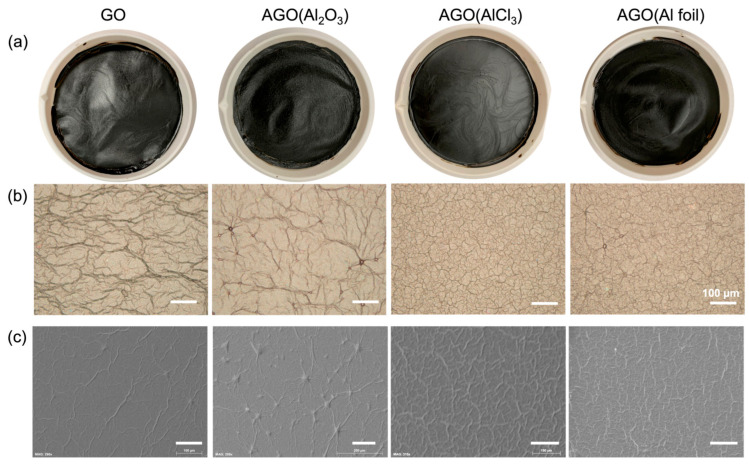
Morphology of the unmodified and Al^3+^-functionalized GO membranes. (**a**) Photographs of the self-assembled GO, AGO (Al_2_O_3_), AGO (AlCl_3_), and AGO (Al foil) membranes in Teflon dishes. OM (**b**) and SEM (**c**) images of the corresponding GO and AGO membranes.

**Figure 3 membranes-12-01237-f003:**
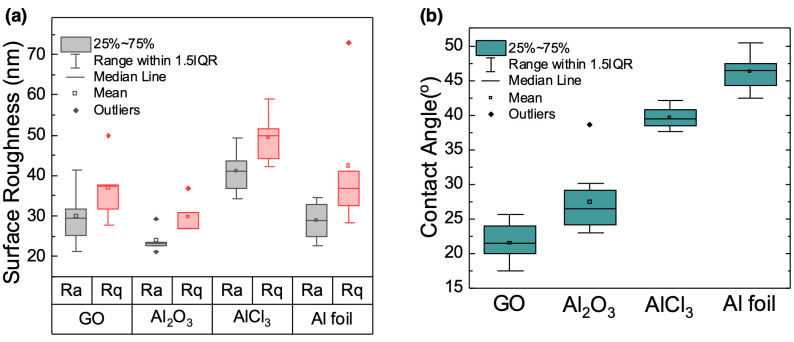
Characterization of the GO and AGO membrane surfaces. (**a**) Average (Ra, gray boxes and dots) and root mean squared (Rq, red boxes and dots) surface roughness values for the unmodified GO, AGO (Al_2_O_3_), AGO (AlCl_3_), and AGO (Al foil) membranes. (**b**) Contact angle measurements of water on the corresponding membranes. The values reported are from 10 trials.

**Figure 4 membranes-12-01237-f004:**
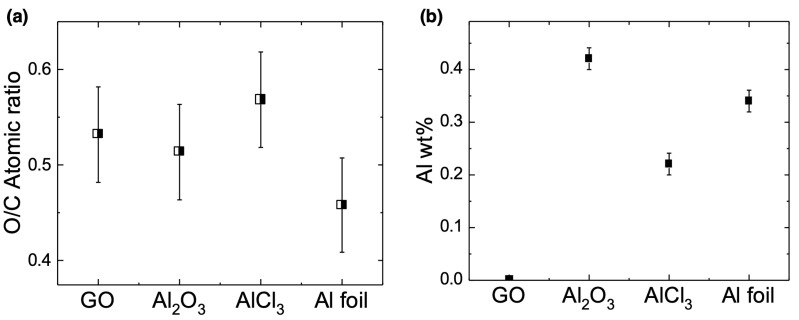
Elemental analysis of the GO and AGO membranes from EDS mapping of the membranes. (**a**) Values for the O/C atomic ratio in the unmodified GO, AGO (Al_2_O_3_), AGO (AlCl_3_) and AGO (Al foil) membranes. (**b**) Weight percent of Al in each membrane.

**Figure 5 membranes-12-01237-f005:**
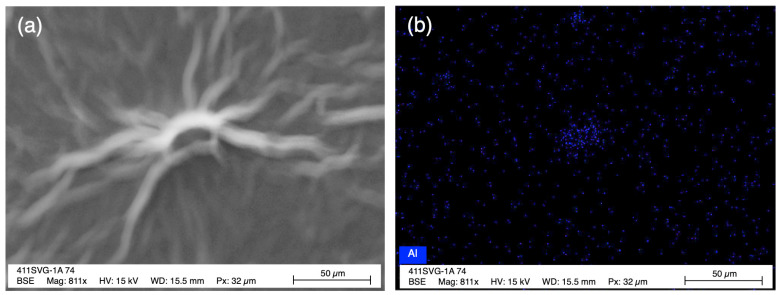
SEM image (**a**) and corresponding aluminum map (**b**) for a particle-like feature on the AGO (Al_2_O_3_) membrane surface.

**Figure 6 membranes-12-01237-f006:**
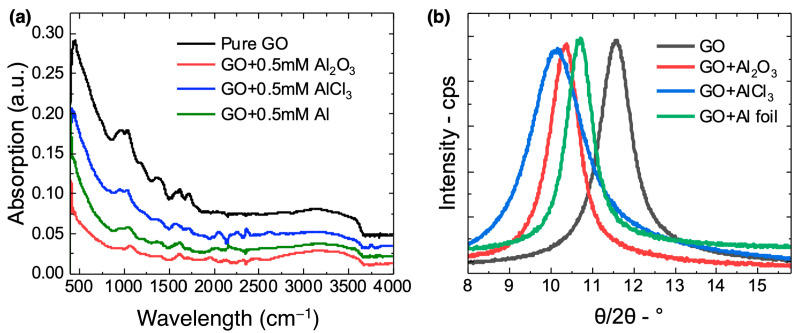
FTIR (**a**) and XRD (**b**) spectra of the unmodified GO, AGO (Al_2_O_3_), AGO (AlCl_3_) and AGO (Al foil) membranes.

**Figure 7 membranes-12-01237-f007:**
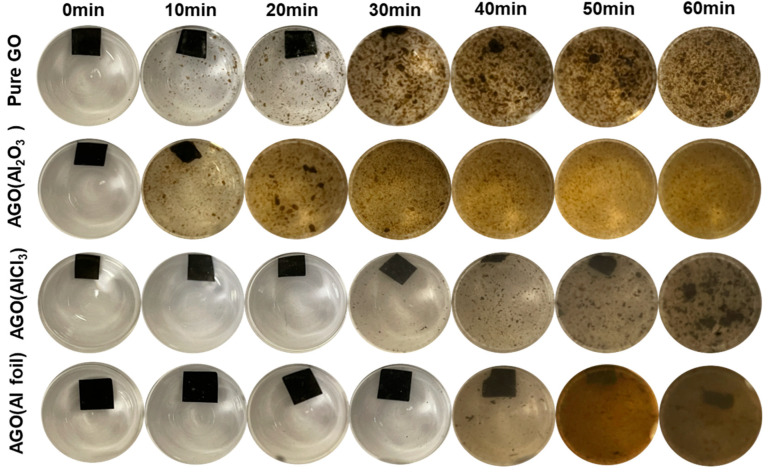
Effect of stirring on the stabilities of the unmodified GO, AGO (Al_2_O_3_), AGO (AlCl_3_) and AGO (Al foil) membranes. Membrane solutions were agitated with a magnetic stir bar spinning at a rate of 200 rpm.

**Figure 8 membranes-12-01237-f008:**
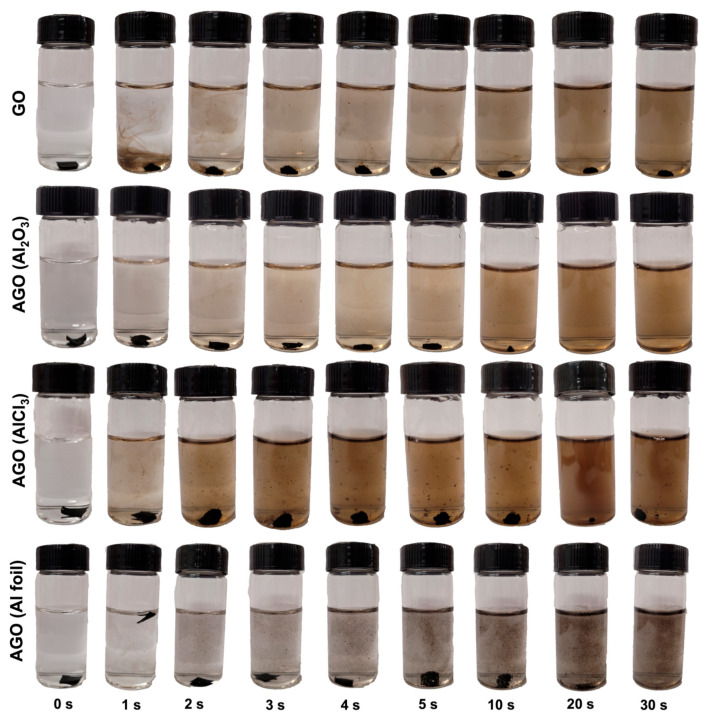
Effect of sonication on the stabilities of the unmodified GO, AGO (Al_2_O_3_), AGO (AlCl_3_) and AGO (Al foil) membranes. Membrane solutions were agitated with a sonication frequency of 60 Hz.

**Figure 9 membranes-12-01237-f009:**
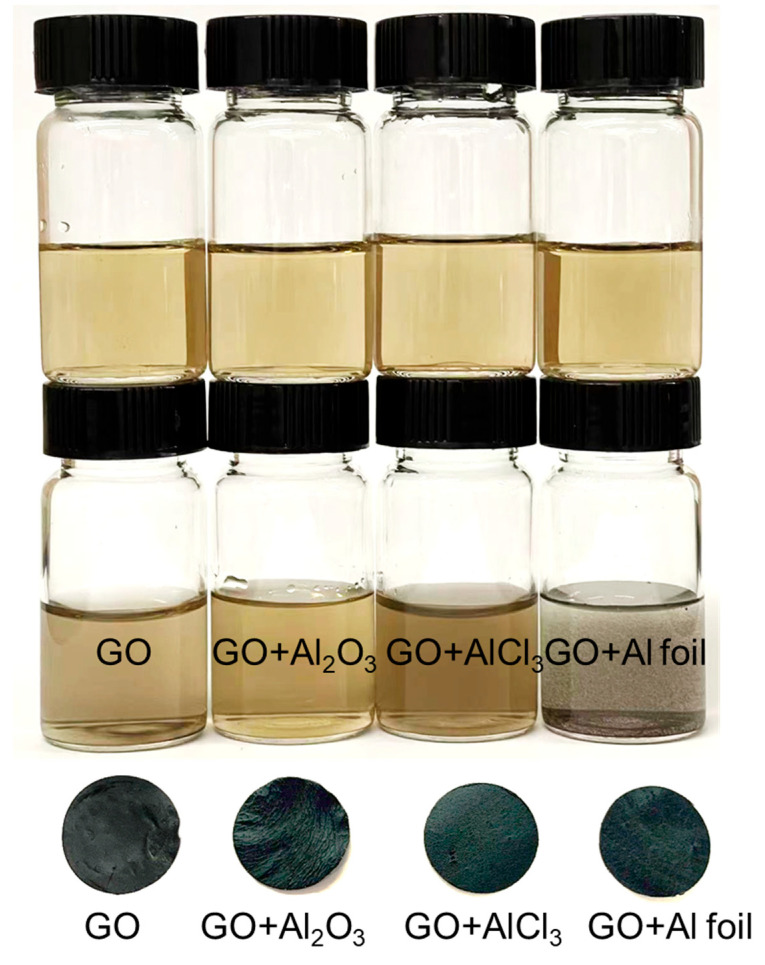
Comparison of the GO sheet solution (top) to sonicated solutions of the GO, AGO (Al_2_O_3_), AGO (AlCl_3_) and AGO (Al foil) membranes (bottom).

## Data Availability

Not applicable.

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
