# Peer review of "Al3+ Modification of Graphene Oxide Membranes: Effect of Al Source"

_membranes, 2022, doi:10.3390/membranes12121237_

Round 1
Reviewer 1 Report
This work showed effect of aluminum (Al) source on the stability of graphene oxide (GO) membrane. Though use of Al to increase stability of GO membrane is well known, authors studied effect of different Al sources on the membrane stability, which will be interesting to scientific community. The manuscript can be accepted after revision. Below are some comments.
1. In introduction section, literature on cation-pi interactions using other cationic molecules (other than metals only) in GO membrane can be added to compare results.
2. The size of all graphs/figures in the manuscript should be increase for better visibility.
3. Interlayer spacing of the samples in wet condition by XRD could be added to compare results.
4. Effect of sonication on stability of membranes could be measured for longer time, 30 sec is short.
5. Why AGO(Al foil) solution become black? Is it converted to reduced GO and poor dispersion?
6. Text on the manuscript is bit longer which can be trimmed to convey same messages.
Author Response
We appreciate the overall positive feedback from the reviewer. We have responded to the reviewer’s comments below:
1. In introduction section, literature on cation-pi interactions using other cationic molecules (other than metals only) in GO membrane can be added to compare results.
The reviewer has made a good point here. Based on this suggestion, we have added a statement to the introduction about how ionic liquids can also be used to increase membrane stability through cation-π interactions. We note, however, that one of the drawbacks of using ionic liquids to form membranes that are water-stable is that the ionic liquid must first be covalently attached to the graphene oxide sheets. Specifically, we have added the following text and references to lines 43-51:
“Previous work has shown that the GO membrane aqueous stability can be enhanced by incorporating molecular species such as ionic liquids into the membrane structure [17-19]. While ionic liquids can promote membrane stability through cation-π interactions, the molecular species must first be covalently attached to the graphene oxide sheet surfaces [17].”
References added:
- Zambare, R.S.; Song, X.; Bhuvana, S.; Tang, C.Y.; Prince, J.S.A.; Nemade, P.R. Ionic Liquid-Reduced Graphene Oxide Membrane with Enhanced Stability for Water Purification. ACS Appl. Mater. Interfaces 2022, 14, 43339–43353.
- Zhao, G.; Zhu, H. Cation–π Interactions in Graphene‐Containing Systems for Water Treatment and Beyond. Adv. Mater. 2020, 32, 1905756.
- Zhou, X.; Zhang, Y.; Huang, Z.; Lu, D.; Zhu, A.; Shi, G. Ionic Liquids Modified Graphene Oxide Composites: A High Efficient Adsorbent for Phthalates from Aqueous Solution. Sci Rep 2016, 6, 38417.\
2. The size of all graphs/figures in the manuscript should be increase for better visibility.
We thank the reviewer for this helpful suggestion. We have increased the sizes of all graphs/figures in the manuscript.
3. Interlayer spacing of the samples in wet condition by XRD could be added to compare results.
The reviewer has suggested an important experiment that can help to demonstrate membrane wettability. We are currently working on these experiments; however, they are proving to be non-trivial, and we believe they are beyond the scope of this manuscript. We plan to include this in-depth XRD study on wet vs. dry GO, AGO, and other metal cation-modified membranes in a future full-length manuscript.
4. Effect of sonication on stability of membranes could be measured for longer time, 30 sec is short.
We agree with the reviewer that 30 seconds is a relatively short time for the sonication experiment. However, we noted that by 30 second the GO and AGO membranes mostly disintegrated and no further changes in the solutions were observed with further sonication. We clarify this point in the manuscript by adding the following sentence and reference to line 786:
“No further changes in the membrane solutions were observed when sonicated past 30 seconds.”
5. Why AGO(Al foil) solution become black? Is it converted to reduced GO and poor dispersion?
The reviewer brings up a good point. This is an interesting question to explore in the future but believe an in-depth study of the solution is beyond the scope of this current work. We currently believe the solution appears to be black not due to the reduction of GO but rather due to poor dispersion of the AGO (Al foil) membrane in water after sonication. The assembled GO membrane is black, whereas the dispersed GO sheet solution is brown. That the sonicated AGO (Al foil) membrane solution appears black rather than brown suggests that there are very few free GO sheets in the solution. We thus believe that the sheets in the sonicated AGO (Al foil) membrane solution remain bound to each other in microscopic membrane pieces, whereas free GO sheets exist in the brown solutions that result from the sonication of the GO, AGO (Al2O3), and AGO (AlCl3) membranes. We clarify our hypothesis concerning this point on line 875-876:
“These black flakes are attributed to AGO sheet aggregates rather than individual reduced GO sheets.”
6. Text on the manuscript is bit longer which can be trimmed to convey same messages.
We appreciate the reviewer noting that the text in the manuscript could be more direct and succinct. We have edited the manuscript such that the word count in the Results and Discussion has been reduced by approximately 500 words.
Reviewer 2 Report
In this paper, the authors advance knowledge on the behavior of GO sheets and membranes when combined with different sources of Al. This is a very well-presented and organized paper. The content is clear and the information is discussed and supported. The figures as of good quality. The Supp Info complement well the information presented.
The length of the paper is of concern, however, it is justified with detailed explanations and supporting citations.
Author Response
We appreciate this overall positive review. We thank the reviewer for noting that the text in the manuscript could be more direct and succinct. We have edited the manuscript such that the word count in the Results and Discussion has been reduced by approximately 500 words.
Reviewer 3 Report
The authors have described the Al3+ modification of GO using different Al sources, and investigated the stability of resultant membranes.
The manuscript is well written with conclusion soundly supported by results.
I have several minor suggestions listed below.
1) In page 9 line 38, the authors assume there is presence of Al2O3 particles. However, there is no evidence. If Al2O3 particles existed, should O/C ratio of Al2O3 modified GO in Figure 4a show highest because Al content in Figure 4b is the highest among others? The authors should reconsider this part. XRD may be a way to identify the existence of Al2O3, authors should present XRD at larger angle.
2) If Al2O3 particles exists, can they be washed away?
3) In the stability discussion, hydrophobicity should also be included because it has significant effect on water stability.
4) Considering permeability and stability, the modification exhibited a sacrifice of permeability to achieve higher stability. This is not ideal for membrane applications. The authors should emphasize more in introduction part regarding the importance of stability.
5) EDS (Figure S3 and S4) and permeability result (Figure S6) are important information. I suggest to move them into the main manuscript.
Author Response
We appreciate the overall positive review. We have addressed the reviewer’s helpful comments below.
- In page 9 line 38, the authors assume there is presence of Al2O3 However, there is no evidence. If Al2O3 particles existed, should O/C ratio of Al2O3 modified GO in Figure 4a show highest because Al content in Figure 4b is the highest among others? The authors should reconsider this part. XRD may be a way to identify the existence of Al2O3, authors should present XRD at larger angle.
The reviewer has made an excellent suggestion. We have collected the recommended XRD spectra of the alumina powder, the unmodified GO membrane, and the AGO (Al2O3) membrane. The alumina powder spectrum shows a small peak near 67°, which is consistent with γ-Al2O3. However, this peak is absent in the spectrum of the AGO (Al2O3) membrane. We do not believe this suggests the absence of alumina particles in the membrane, rather, we believe there are not enough alumina particles within the membrane to produce a strong signal. We are currently working on experiments to optimize the XRD signal at larger angles. Moreover, we cannot use the O/C ratio to confirm the presence of the particles, as water may contribute to the oxygen content of all membranes studied. Our strongest pieces of evidence for the presence of alumina particles within the AGO (Al2O3) membrane are the SEM images and corresponding aluminum elemental maps, which we have moved from the SI to the main text of the manuscript (now Figure 5). Here, the density of aluminum is much higher near the particle-like feature on the membrane surface than it is on other parts of the membrane.
- If Al2O3 particles exists, can they be washed away?
The reviewer raises an interesting question. Based on the OM and SEM images of the AGO (Al2O3) membrane, we believe that some the alumina particles are embedded within the membrane. Because these AGO (Al2O3) membranes are particularly fragile and unstable in water under stirring, we need to develop a protocol to wash the membranes without damaging them. This is proving to be non-trivial and beyond the scope of this study. Once we develop the protocol, we plan to include the results in a full-length manuscript that includes the XRD spectra of GO, AGO, and other metal cation functionalized membranes at higher angles before, during, and after membrane washing. These data will be also used to better understand the wettability, stability, and permeability of GO membranes functionalized with aluminum as well as other cations that we are currently studying.
- In the stability discussion, hydrophobicity should also be included because it has significant effect on water stability.
We thank the reviewer for this helpful comment. We agree that membrane hydrophobicity greatly affects water stability. We have clarified this point by adding the following statements:
“Our previous work indicated that an increase in membrane stability in water was related to an increase in membrane hydrophobicity [22]. Here, as indicated by the contact angle results, all AGO membranes were more hydrophobic than the unmodified GO membrane, but only the AGO (AlCl3) and AGO (Al foil) membranes were more stable in water than the unmodified GO membrane.” – Line 931 - 935
“Even though these positively charged particles can screen the negative charges on the AGO (Al2O3) membrane, they likely interrupted van der Waals interactions that stabilize unmodified GO membranes in water without providing the stabilizing effects of free Al3+ cations.” – Line 955 - 958
“While these different interactions resulted in an increase in membrane hydrophobicity and consequently water stability, they likely affected the membrane structure and thus the mechanism by which each membrane disintegrated in water under mechanical stresses.” – Line 965 - 968
- Considering permeability and stability, the modification exhibited a sacrifice of permeability to achieve higher stability. This is not ideal for membrane applications. The authors should emphasize more in introduction part regarding the importance of stability.
We agree with the reviewer that an increase in membrane stability at the expense of a decrease in membrane water permeability is not ideal for graphene oxide membrane-based filters. However, if graphene-oxide membrane-based filters are to be reused, it is imperative that these membranes maintain their structural integrity upon washing and recycling. Thus, we believe that a small reduction in membrane permeability that results from creating a highly robust and recyclable filter is an acceptable consequence. We have further noted the importance of membrane stability in water filtration applications by adding the following text and corresponding reference to lines 38 - 41 of the introduction:
“In order to be useful materials in water filtration and separation technologies, the membranes must maintain their structural integrity in water, which is especially important if GO membrane are to be recycled and reused for multiple filtrations [16].”
References added:
- Xing, C.; Han, J.; Pei, X.; Zhang, Y.; He, J.; Huang, R.; Li, S.; Liu, C.; Lai, C.; Shen, L.; et al. Tunable Graphene Oxide Nanofiltration Membrane for Effective Dye/Salt Separation and Desalination. ACS Appl. Mater. Interfaces 2021, 13, 55339–55348.
We have reiterated this point when discussing the reduced permeability of our AGO membranes by adding the following text to lines 949 - 951:
“Even though enhanced membrane permeability is important for filtration applications, the enhanced water stability seen in our AGO (AlCl3) and especially AGO (Al foil) membranes may allow them to serve as recyclable and reusable filters.”
- EDS (Figure S3 and S4) and permeability result (Figure S6) are important information. I suggest to move them into the main manuscript.
We thank the reviewer for acknowledging the importance of these results. However, we believe adding Figures S3 and S6 and their corresponding information to the manuscript will significantly increase its length. We have moved Figure S4 to the main text (now Figure 5) because it supports the hypothesis that the particles on the AGO (Al2O3) membrane are due to unreacted particles and will not significantly increase the length of the manuscript.
Round 2
Reviewer 1 Report
Authors revised manuscript based on previous comments, however manuscript can be accepted after revision based on below minor comments.
1. Investigation of interlayer spacing in wet condition is important in layered materials and should be included in the manuscript.
2. Is there leaching of Al from the membrane when it exposed long time in aqueous environment? Experimental results on leaching of Al should be added in the manuscript.
Author Response
We greatly appreciate the important suggestions given by the reviewer. We agree with the reviewer that a thorough characterization of the membrane properties after soaking in water are important to understanding how they can be effectively used in water-based applications like filtration. However, the studies suggested by the reviewer are non-trivial and will likely take several weeks to complete. We are currently in the process of preparing a full-length manuscript on the properties of graphene oxide membranes that are stabilized with different divalent and trivalent cations. For this future manuscript, we will include studies on the d-spacing for wet vs. dry membranes modified with different cations along with studies on the leaching of cations from these different membranes after soaking in water for long periods of time.
While we have not yet completed studies on the leaching of ions from cation-modified membranes, we have found evidence in the literature for minimal leaching of Al3+ from AGO membranes. We have thus included the following text to the introduction (Line 61):
“Previous studies have shown that AGO membranes can persist in water for several weeks, leaching GO sheets to an extent less than 0.1% [11]. This enhanced stability indicates strong interactions between the GO sheets and Al3+ cations that likely minimize leaching of the GO sheets and Al3+ cations from the membrane.”